# Use of Sonophoresis with Corticosteroids in Carpal Tunnel Syndrome: Systematic Review and Meta-Analysis

**DOI:** 10.3390/jpm12071160

**Published:** 2022-07-17

**Authors:** Francisco Javier Martin-Vega, Maria Jesus Vinolo-Gil, Veronica Perez-Cabezas, Manuel Rodríguez-Huguet, Cristina Garcia-Munoz, Gloria Gonzalez Medina

**Affiliations:** 1Department of Nursing and Physiotherapy, University of Cadiz, 11009 Cadiz, Spain; javier.martin@uca.es (F.J.M.-V.); veronica.perezcabezas@uca.es (V.P.-C.); manuel.rodriguez@uca.es (M.R.-H.); cristina.garciamunoz@uca.es (C.G.-M.); gloriagonzalez.medina@uca.es (G.G.M.); 2Rehabilitation Clinical Management Unit, Interlevels-Intercenters Hospital Puerta del Mar, Hospital Puerto Real, Cadiz Bay-La Janda Health District, 11006 Cadiz, Spain

**Keywords:** phonophoresis, sonophoresis, carpal tunnel syndrome, median neuropathy

## Abstract

Carpal tunnel syndrome is a neuropathic disease. It is one of the most frequent musculoskeletal pathologies affecting the upper limbs. One of most frequently used non-surgical treatments is corticosteorids. There are several alternatives for corticosteroids administration. One of them is phonophoresis, this being an effective and painless method of treatment. A systematic review and meta-analysis have been conducted over the use of phonophoresis with corticosteroids for the treatment of carpal tunnel syndrome compared to other non-surgical treatment methods. Keywords from Medical Subjects Headings (MeSH) were used in the following databases: Wos, Scopus, CINHAL, SciELO and PeDro. A total of 222 potentially relevant articles were retrieved. Eleven articles analysing the efficacy of phonophoresis with corticosteroids in reducing pain symptoms in individuals with carpal tunnel syndrome were included, 10 of which were used to conduct the meta-analysis. A conclusion could not be reached as to the application of phonophoresis with corticosteroids being better than other treatment methods, except for the perception of pain and an improved motor and sensory nerve conduction in cases of mild to moderate carpal tunnel syndrome.

## 1. Introduction

Carpal tunnel syndrome (CTS) is a neuropathic disease that occurs at the wrist area when the median nerve is affected. It is one of the most frequent musculoskeletal pathologies affecting the upper limb [1]. It features a high prevalence among the working force and it reaches a rate in the general population of 276/100,000 individuals [2]. Symptoms may include a sensation of tingling, numbness and paresthesia, as well as pain and weakness in the area of the hand innervated by the median nerve [3]. Surgical treatment is reserved for severe cases or when conservative treatment is not successful [4]. One of most frequently used non-surgical treatments is corticosteorids injections [5]. Due to the direct inoculation into the injury, it is highly effective in mild cases and in the short term, but it shows arguable results in the long term [6]. There are several alternatives to corticosteroids administration. One of them is phonophoresis, this being an effective and painless method of administration [7]. It is applied by using the cavitation effect produced by an ultrasonic wave allowing drug penetration [8]. This method of administration has been used since 1950. From then on, its functional characteristics have been modified and improved [9]. The advantages observed with this method of treatment include less harmful effects on organs, due to a reduced systemic concentration of drugs, reduced side effects, and an improved adherence of patients to therapy, since it is a painless technique [10,11].

In spite of the systematic reviews that exist on this subject [12,13,14,15], no meta-analysis focused on phonophoresis with corticosteroids compared to other non-surgical techniques for carpal tunnel syndrome has been found. Therefore, the aim of this systematic review and meta-analysis is to conduct a systematic review and meta-analysis over the use of phonophoresis with corticosteroids for the treatment of carpal tunnel syndrome compared to other non-surgical treatment methods.

## 2. Materials and Methods

### 2.1. Design

A systematic review and meta-analysis was conducted and recorded in PROSPERO (CRD42022338100) using the Preferred Reporting Items for Systematic Reviews and Meta-analysis (PRISMA) guidelines [16]. Search was conducted between September 2021 and April 2022.

### 2.2. Search Strategy

Keywords from Medical Subjects Headings (MeSH) were used in the following databases: Web of Science (WoS), Expertly curated abstract & citation database (Scopus), PubMed database, Cumulative Index to Nursing & Allied Health Literature (CINHAL), Scientific Electronic Library Online (SciELO) and Physiotherapy Evidence Database (PEDro). The following terms combined with Boolean operators were used: “Phonophoresis”, “Sonophoresis”, “carpal tunnel syndrome”, “Median Neuropathy” and “Treatment(s)”. This search was restricted to controlled clinical trials. Filters were not applied when searching (Table 1).

### 2.3. Elegibility Criteria

The eligibility criteria were established based on the PICO [17] model: (P) Population—subjects diagnosed with carpal tunnel syndrome; (I) Intervention—treatment consisting in phonophoresis with corticosteroids. (C) Comparator—sham and other treatments; (O) Outcomes—related to pain and/or nerve conduction. Studies not being clinical trials or conducted on humans were excluded. No language or date restriction were used.

### 2.4. Studies Selection and Data Abstraction

Two reviewers (F.J.M.V. and G.G.M.) conducted the studies selection process, eliminated duplicates, and reviewed and systematically abstracted data. A third reviewer (V.P.C.) helped to reach an agreement in case of dispute. For this, the Mendeley reference manager version 1.19.8.0 was used.

Data abstracted from each of the studies included were: (i) authors and publication date; (ii) subjects’ related data (number of subjects, sex, age, number of bilateral cases, study groups and degree of injury); (iii) data related to assessment (measuring tools used, number and timing of assessments); (iv) data related to the intervention with phonophoresis (type of intervention, type and quantity of active ingredient used); (v) parameters of techniques used, duration and frequency of the intervention; (vi) achieved outcomes.

### 2.5. Assessment of the Methodological Quality of the Studies

For methodological assessment purposes, the PEDro [18] scale was used. This scale comprises 11 items related to selection, performance, detection, information, and allocation. Each item scores one point if the studies meet the criteria, except item 1, which is not used for final calculation purposes. A score higher than or equal to 6 is considered a level of evidence 1 (10–9: excellent; 8–6: good) and a score lower than or equal to 5 is considered a level of evidence 2 (5–4: fair; below 4: poor) [19].

### 2.6. Statistical Analysis

The IT tool Review Manager (RevMan) was used to conduct the meta-analysis. Version 5.4. The Cochrane Collaboration, 2020 [20]. Data were gathered by variable and form of drug administration. Generic inverse variance was applied for the analysis of each sub-group. When grouping could not be done, mean difference was applied. When I^2^ value was above 50% the random effect model was used. The effect size was obtained through the Z value in the Test for overall effect and its p value (*p* < 0.05 significant; *p* > 0.05 non-significant). Sensitivity analysis has been conducted for all studies included in all the variables subject to analysis. To verify the homogeneity of studies, the value for *p* of Heterogeneity (*p* < 0.05 Homogeneity; *p* > 0.05 Heterogeneity) was used. The risk of bias was estimated through the Begg and Egger tests, with the epidemiological data analysis programme (EPIDAT) [21], in addition to the funnel plots outcomes.

## 3. Results

A total of 222 potentially relevant articles were retrieved (Figure 1). Eleven articles analysing the efficacy of phonophoresis with corticosteroids in reducing pain symptoms in individuals with carpal tunnel syndrome were included, 10 of which were used to conduct the meta-analysis (Figure 1).

The total sample included 434 subjects. The average was 39.5 subjects (a bracket between 31 and 54 subjects). Only four studies [22,23,24,25] identified the prevalence of women, but not the others. The age range of subjects was 39–54 years. The bilateral pathology is present in a range of 22.5% and 44.6% of subjects, except for four studies which do not reflect it [22,25,26,27]. Carpal tunnel syndrome has been diagnosed at a mild to moderate level, except for five studies which do not refer to it [25,27,28,29,30].

In relation to the comparison of different alternative therapies, treatment based on phonophoresis with corticosteroids was compared to local corticosteroid injection [22,23,24,25,29], iontophoresis with corticosteroids [24,25,26,27,31], phonophoresis with nonsteroidal anti-inflammatory drugs [28,29,32], the application of ultrasounds with contact gel [30,32], the application of ultrasounds simulation [30] and the use of a splint mainly at night [27,28,29].

Concerning the tools used for diagnosis purposes, all of the studies included the Boston Carpal Tunnel Questionnaire (BCTQ) scale [33] which consists of a 19-item self-administered questionnaire with two subscales: a subscale measuring symptom severity (11 items) and a scale assessing functional status (eight items) and/or nerve conduction studies (NCS) on the median nerve [34] where major motor latency (LMMotor) is analysed, sensory major latency (LMSensor), sensory nerve action potential amplitude (SNAPam), motor nerve action potential amplitude (CMAPam), sensory nerve conduction velocity (CNVS) and motor nerve conduction velocity (CNVM) are measured. Some of them used the visual analogue scale (VAS) for patient self-assessment of pain sensation [35], as well as the dynamometer for measuring grip strength [23,25,26,27,29,31] and pinch strength [23,25,26,31], evaluation of manual dexterity [23,27,29], Phalen and Tinel’s tests [28,29], degree of paresthesia [23,26], ecographic analysis of the cross-sectional area of the median nerve [28], as well as the Semmes-Weinstein test and the Duruoz Hand Index [29].

The assessment of different diagnosis tools was conducted before and after the intervention. In some studies a subsequent evaluation after four weeks [26,31], two months [24], and three months [23,24,25,27,28,29,30] was included.

Steroideal active ingredients enhancing phonophoresis efficacy compared to other treatment methods were: betamethasone valerate 0.1% [28,29] compared to phonophoresis with diclofenac diethylammonium and local injection with betamethasone dipropionate. Betamethasone 0.1% [27] compared to the use of a splint. Dexamethasone sodium phosphate 0.4% [26,31] compared to its application with iontophoresis, although Aygül R et al. [24] do not find significant variations using the same active ingredient and comparing both methods, except in the case of using a lower drug concentration (0.1%). Dexamethasone 0.1% [30] compared to ultrasounds with contact gel.

The treatment was applied for four weeks [22,32], or during three weeks [24,25,27,28,29] or two weeks [23,26,30,31]. The frequency of sessions ranged from one session a day to three sessions a week.

With regard to the parameters used for the application of phonophoresis with corticosteroids, most of the studies [22,25,26,27,31,32] applied a 1 MHz frequency, except for three of them which used 3 MHz [24,28,29], and two studies which do not state it [23,30]. With regard to the emission mode, the continuous mode [25,27,32] and the pulsed mode [26,31] are used or it is not specified [22,23,24,28,29,30]. The intensity applied was 1 W/cm^2^ in all of the studies except in two of them in which it reached 1.5 W/cm^2^ [28,29] and another one applying 0.1 W/cm^2^ [30]. The time of emission was 10 min per session in most studies, except for four of them in which the session took five minutes [22,26,30,31]. Table 2 shows the main characteristics of the studies included in this systematic review.

In relation to quality from a methodological perspective, the average resulting from all the studies is considered as good (7/10) (Table 3).

### Study Groups Included in the Meta-Analysis

Twelve meta-analyses grouped by variable were made (Figure 2, Figure 3, Figure 4, Figure 5, Figure 6, Figure 7, Figure 8, Figure 9, Figure 10, Figure 11, Figure 12 and Figure 13).

These results show that no statistically significant effect arises from phonophoresis in any of the analysed variables. In the case of variables such as Pain (−1.5, 0.15), LMMotor (−0.51, 0.16) and LMSensor (−0.81, 0.01), there is a positive trend for phonophoresis with certain values remarkably close to significance.

The sensitivity analysis for all the studies in every analyzed variable does not significantly modify the outcomes, therefore, our analysis is solid. As for LMSensor, when removing the study by Aygül, R. et al. I, the outcome changes in favour of phonophoresis. However, it should be noted that this outcome was already remarkably close to significance (−0.81, 0.01).

Studies analysed for the variables pain, BCTQ Sensorial, BCTQ Total, LMMotor, LMSensor, Grip strength and CNVS were considered homogeneous. The rest of the studies for the remaining variables were heterogeneous (Figure 14, Figure 15, Figure 16, Figure 17, Figure 18, Figure 19, Figure 20, Figure 21, Figure 22, Figure 23, Figure 24 and Figure 25).

## 4. Discussion

The prevalence of women observed during this review, as well as the subjects’ range age, is in line with the epidemiological studies carried out in the past [36,37,38,39]. The presence of a higher quantity of women may be due to their prevalence in types of employment prone to suffer from this pathology [39]. Furthermore, the age range stated does not imply that this pathology cannot be found in other age groups [40].

Regarding the bilateral occurrence of the pathology, the average ratio resulting from this review is slightly lower than that stated in other searches [37,39]. This may be due to having excluded this profile of patients in certain studies, whether by exclusively selecting the dominant hand [27] or by randomly selecting one of the hands [22].

Although all of the studies included in this review state the efficacy of phonophoresis when applied with corticosteroids in carpal tunnel syndrome, a consensus has not been reached as to its therapeutical preeminence compared to other types of treatment.

Therefore, when comparing the application of phonophoresis with corticosteroids to other methods, this is not decisive concerning an improvement of the studied variables. When compared to injections, it is observed that electrophysiologic parameters and pain improve with injections, but this is not the case when applying phonophoresis. This is in contrast with Soyupek F et al. [29], who observed improved physiologic parameters but not pain relief.

When comparing this application to iontophoresis, it is observed that phonophoresis provides higher benefits [31], and more particularly to the electrophysiologic parameters [26]. This contradicts other studies [24,27], in which no statistically significant differences are found between both methods. Phonophoresis features higher or equivalent efficacy when compared to the application of splints [27,28,29] or low level laser [22], respectively.

Clinical symptoms and functionality improve both after the application of corticosteroids via phonophoresis and the application of phonophoresis without any drug.

Concerning evaluation tools, much heterogeneity exists. The reason may be found in the lack of a standardised protocol for this purpose [1,41]. Three methods were the most frequently used out of the different methods described: (1) the Boston Carpal Tunnel Questionnaire [33] scale due to its high reliability and validity, as well as its ability to be rendered and adapted to other languages [42,43,44,45,46]; (2) the electrophysiologic study on the velocity of the median nerve conduction according to the American Association of Electro-diagnostic Medicine (AAEM) guidelines [34]. While its proved objective validity, high specificity and sensitivity [47], some authors recommend to use it in addition to other instruments [5,48]. This is due to potential false negatives in a ratio ranging from 16 to 34% [1]; and (3) the visual analogue Scale, which is highly recommended for measuring pain [49]. The reason why these tests are more frequently observed may be due to the poor diagnostic utility of other evaluation tools as described by Li Pi Shan R et al. [50], despite the fact that the rest of the diagnostic instruments used in the studies have been to a lower or greater extent credited in other research activities [51,52,53,54,55,56,57].

As for the active ingredients, there is also a great heterogeneity. Anti-inflammatory and analgesic actions of dexamethasone valerate have been already described in other studies [58,59]. When comparing the use of nonsteroidal anti-inflammatory drugs (NSAIDs) to corticosteroids, Soyupek F et al. [28,29] describe a higher efficacy of dexamethasone valerate compared to diclofenac, which is in line with the studies conducted by Iannitti et al. [59].

Bakhtiary AH et al. [26,31] describe a higher efficacy of dexamethasone sodium phosphate when applied with phonophoresis than iontophoresis, in contrast with Aygül R et al. [24], who do not observe any significant difference. This may be due to the latter using a lower concentration of the active ingredient. However, the studies conducted by Akinbo SR et al. [60] did not find significant differences either when comparing both methods applied to knee arthritis using this same active ingredient at similar ratios to those applied in the studies by Bakhtiary AH et al.

With regard to the parameters used in phonophoresis, our outcomes do not allow us recommend the use of 1 MHz frequency instead of 3 MHz [12]. This may be due to the penetration efficacy mostly depending on the amount of energy applied and the time of application [61]. Even so, the use of high frequencies is justified, as they increase the density of the energy necessary to improve skin permeability [62]. Nonetheless, frequencies lower than 1 MHz allow deeper drug penetration, this being relevant for transdermal drug administration [9]. It is likely that no differences are observed in relation to the use of various intensities (W/cm^2^), which is in line with the outcomes observed during a systematic review conducted by Huisstede BM et al. [12].

The meta-analysis proves that, although no statistically significant relation is established between the use of phonophoresis compared to other kinds of drug administration, there is indeed a significant trend in the case of pain, LMMotor and LMSensor. This suggests that using phonophoresis may be beneficial in the case of patients suffering from pain or for whom nerve conduction, both motor and sensory, is altered.

No other systematic reviews or meta-analysis have been found for comparing our outcomes.

### Review Strengths and Limitations

A systematic review and a meta-analysis have been conducted by following a specific methodology. Sensitivity analysis shows stability for the final measure obtained. All the studies found have been reviewed without using any filters. Clinical trials have been used showing a medium-high methodological quality.

The literal translation of studies drafted in Turkish or Persian could indicate a general or biased overview of the global construction the authors wished to convey.

A differentiation in syndrome severity could not be realised, as certain authors did not state it [25,27,28,29,30].

For the analysis of some variables, the number of studies were scarce. Although variables could be categorised into groups, it was observed that some of them were measured on different nerves, which could have an impact on extremely specific assessments. We recommend the use of updated clinical practice guidelines as far as the electrophysiologic studies is concerned [34], and a large consensus on the most relevant parameters to be asessed.

## 5. Conclusions

The application of corticosteroids by phonophoresis do not seem better that other treatments according with the results, except for the perception of pain and an improved motor and sensory nerve conduction in case of mild to moderate carpal tunnel syndrome.

Concerning the evaluation tools, for the active ingredients and phonophoresis parameters, much heterogeneity does exist. For future studies we recommend these variables to be homogenised in order to achieve more conclusive outcomes and parameters to conform to the treatment objectives.

## Figures and Tables

**Figure 1 jpm-12-01160-f001:**
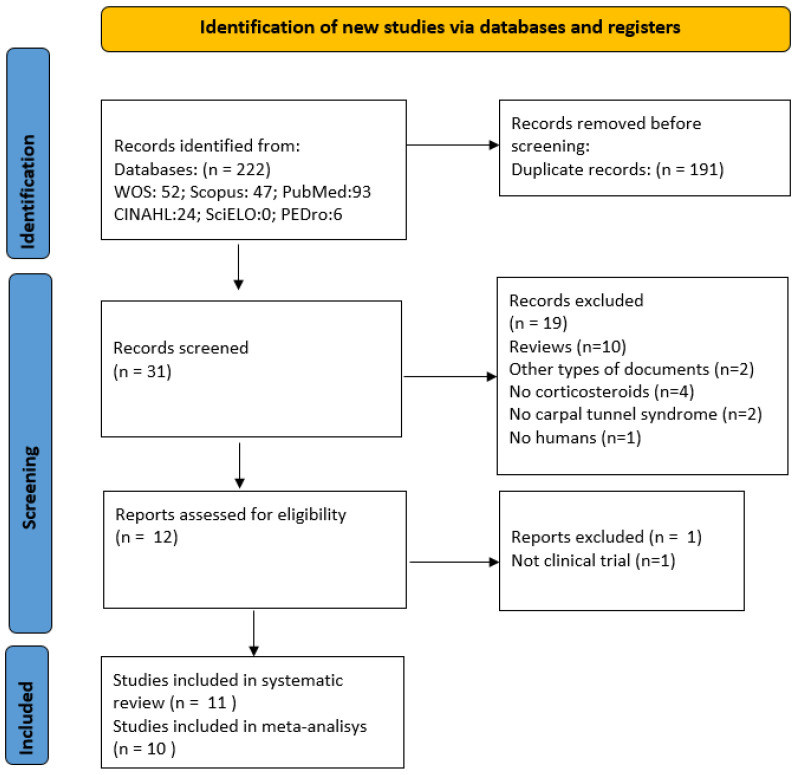
Flowchart of the studies selection process.

**Figure 2 jpm-12-01160-f002:**
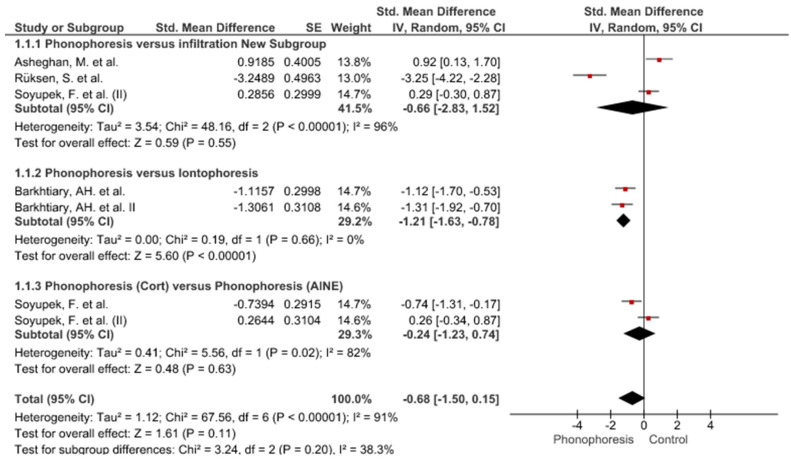
Phonophoresis for pain.

**Figure 3 jpm-12-01160-f003:**
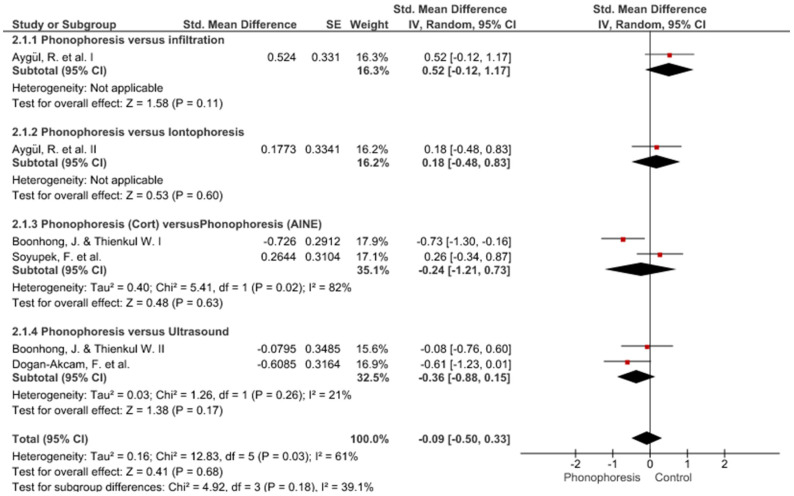
Phonophoresis for BCTQ Sensorial.

**Figure 4 jpm-12-01160-f004:**
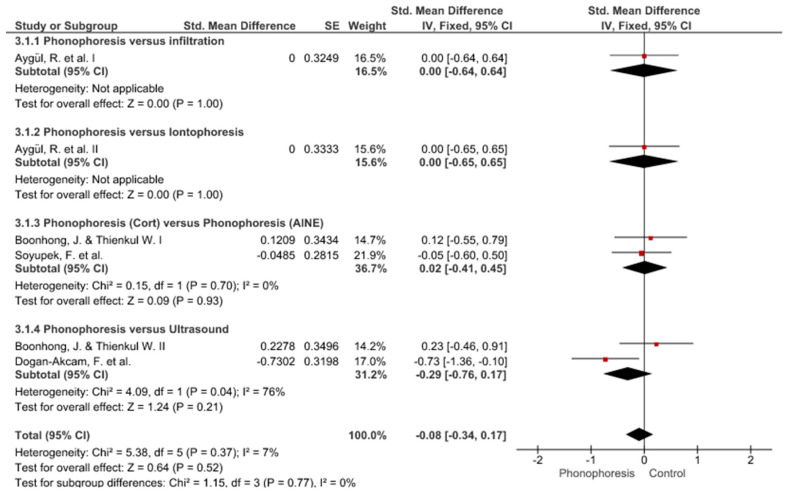
Phonophoresis for BCTQ Function.

**Figure 5 jpm-12-01160-f005:**
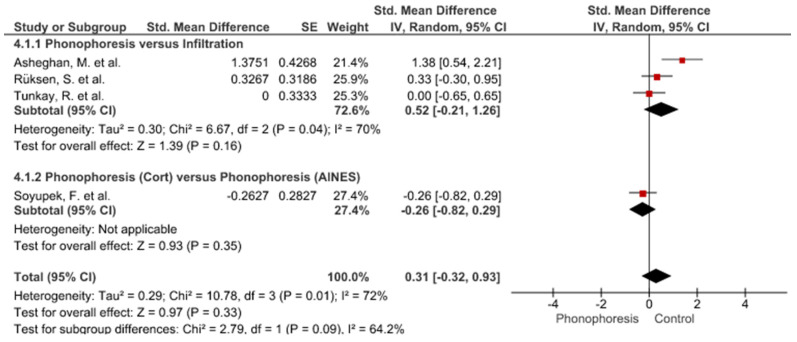
Phonophoresis for BCTQ Total.

**Figure 6 jpm-12-01160-f006:**
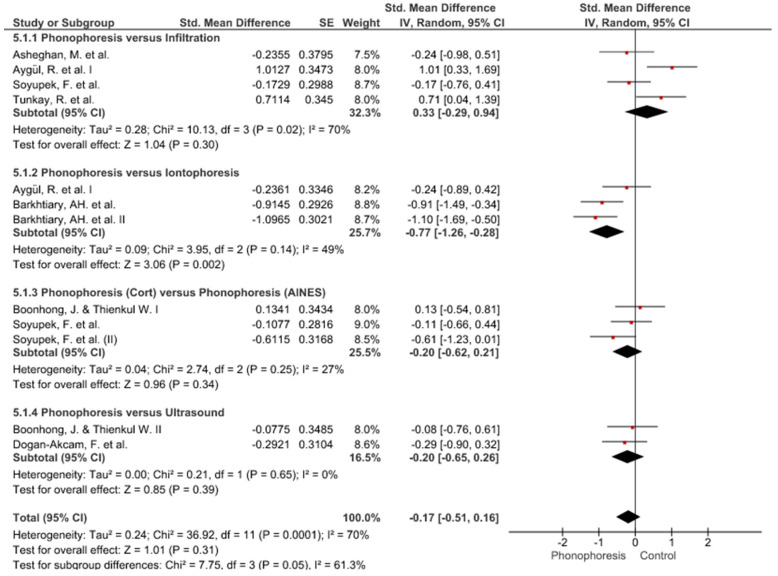
Phonophoresis for LMMotor.

**Figure 7 jpm-12-01160-f007:**
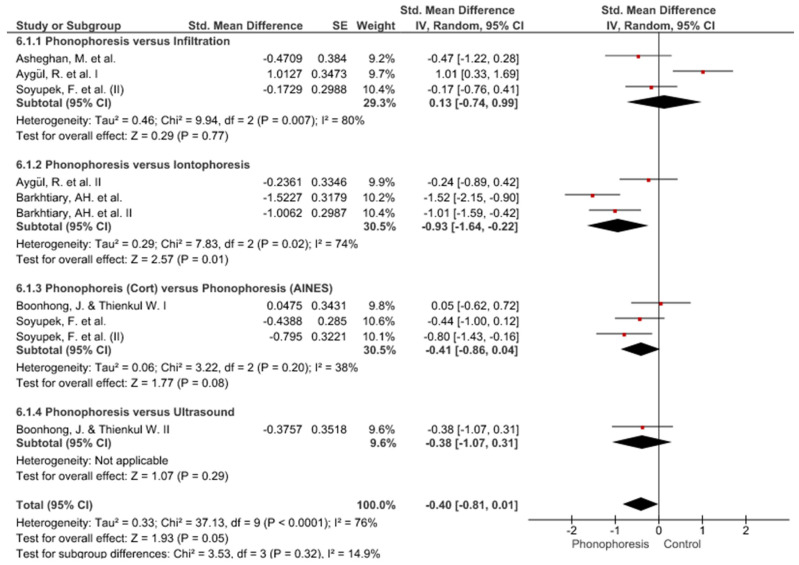
Phonophoresis for LMSensor.

**Figure 8 jpm-12-01160-f008:**
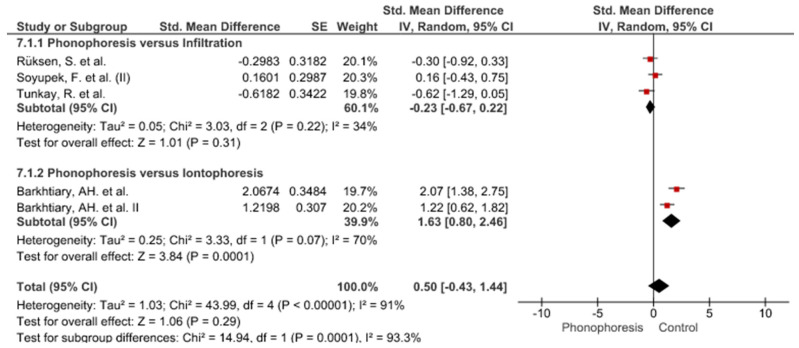
Phonophoresis for Grip strength.

**Figure 9 jpm-12-01160-f009:**
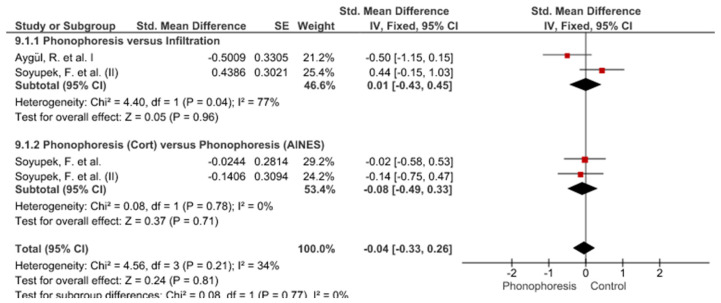
Phonophoresis for SNAPam.

**Figure 10 jpm-12-01160-f010:**
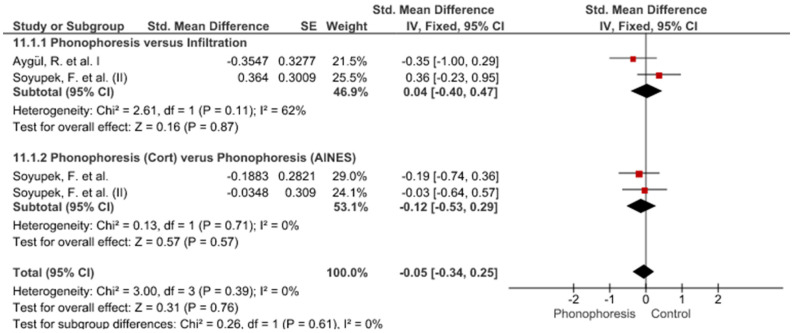
Phonophoresis for CNVM.

**Figure 11 jpm-12-01160-f011:**
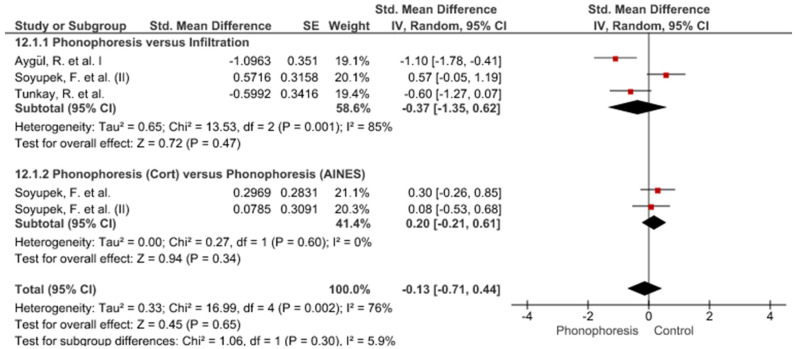
Phonophoresis for CNVS.

**Figure 12 jpm-12-01160-f012:**
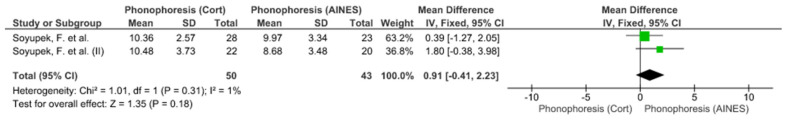
Phonophoresis for CMAPam.

**Figure 13 jpm-12-01160-f013:**
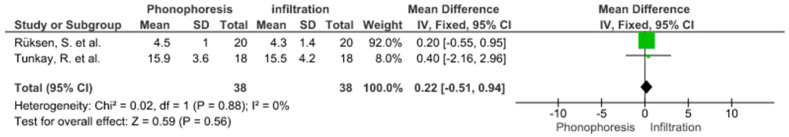
Phonophoresis for Pinchmeter.

**Figure 14 jpm-12-01160-f014:**
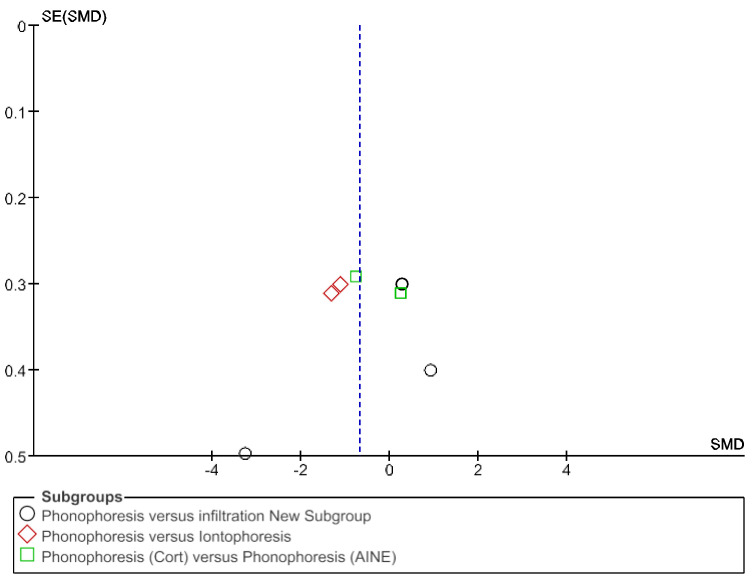
Phonophoresis for pain.

**Figure 15 jpm-12-01160-f015:**
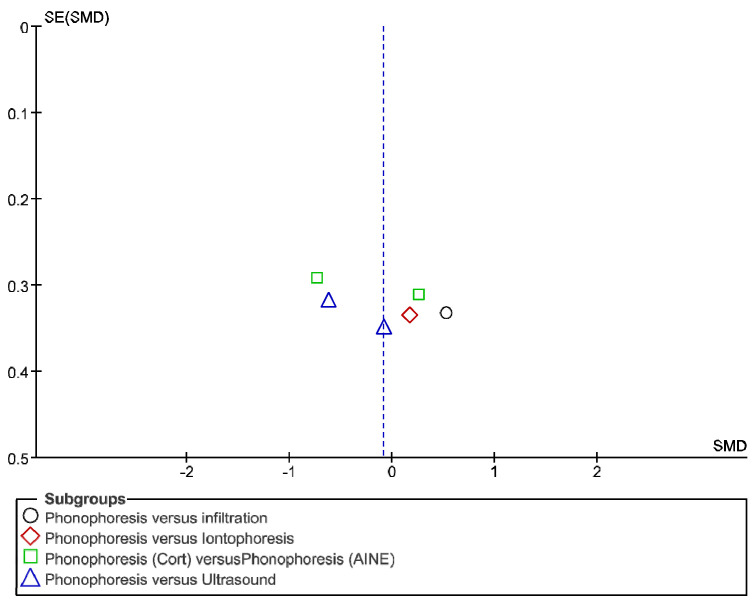
Phonophoresis for BCTQ Sensorial.

**Figure 16 jpm-12-01160-f016:**
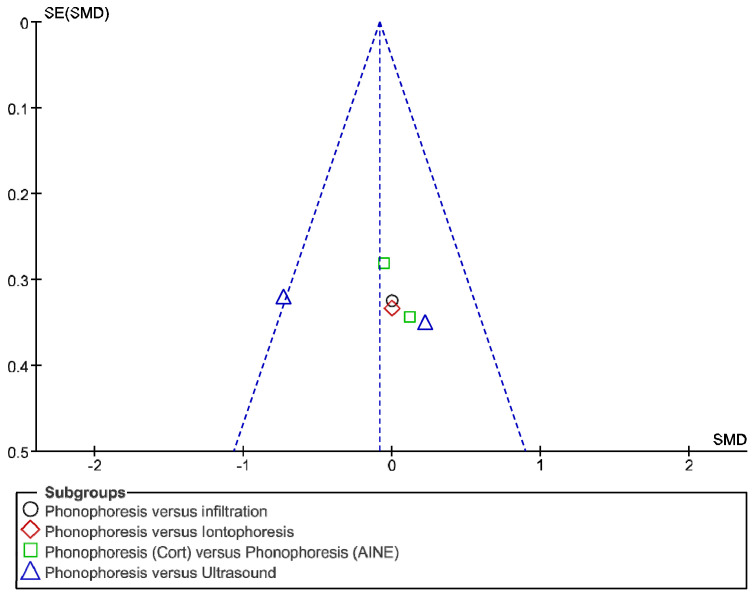
Phonophoresis for BCTQ Function.

**Figure 17 jpm-12-01160-f017:**
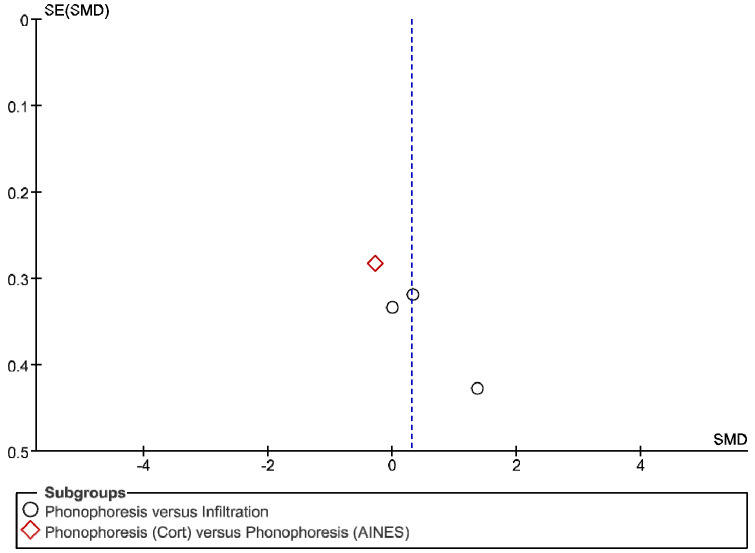
Phonophoresis for BCTQ Total.

**Figure 18 jpm-12-01160-f018:**
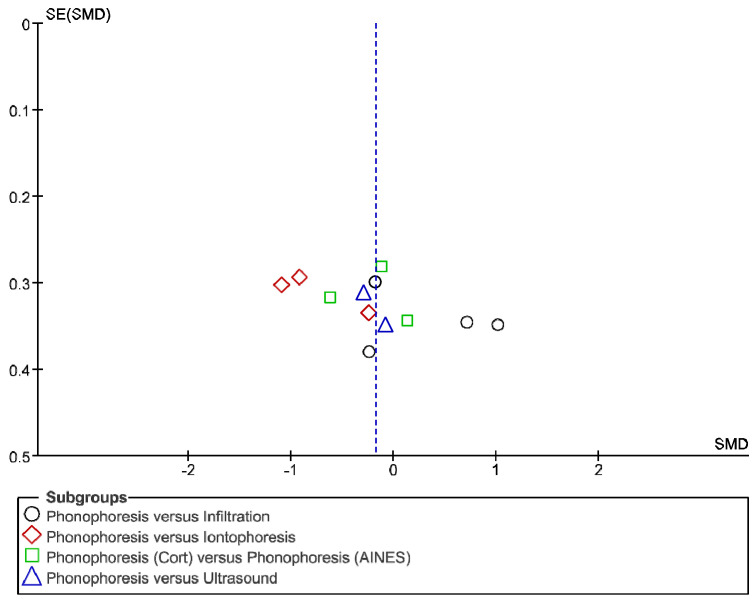
Phonophoresis for LMMotor.

**Figure 19 jpm-12-01160-f019:**
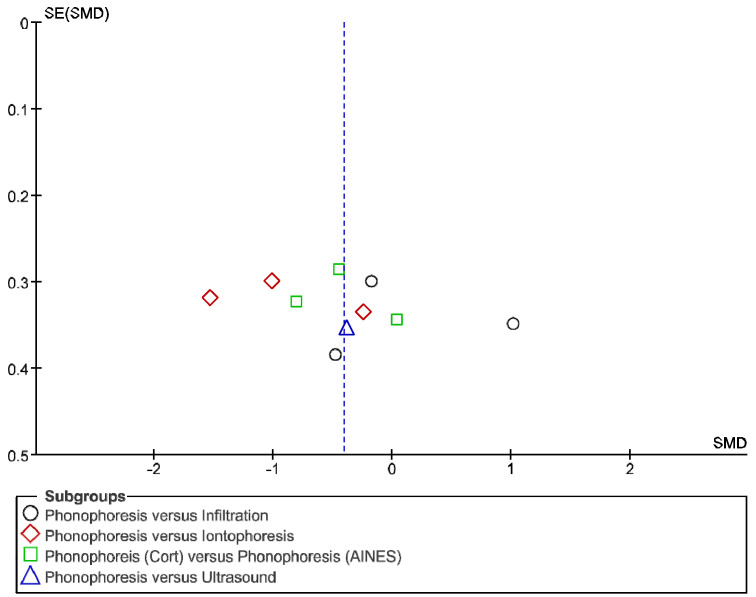
Phonophoresis for LMSensor.

**Figure 20 jpm-12-01160-f020:**
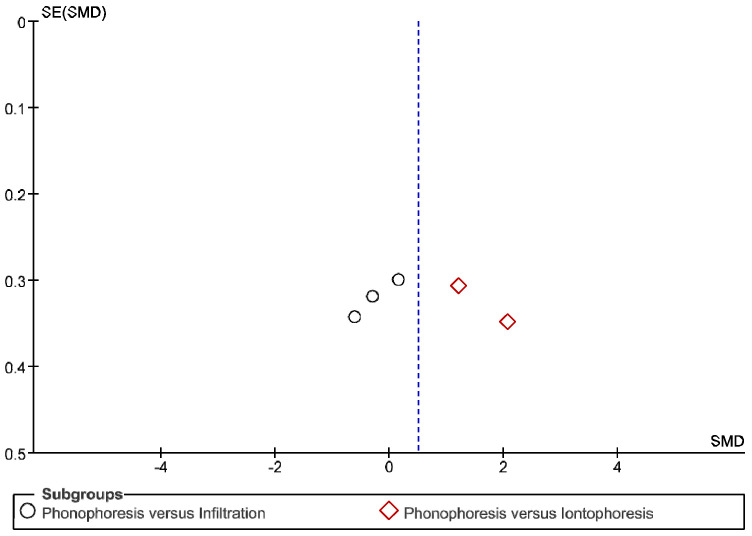
Phonophoresis for Grip strength.

**Figure 21 jpm-12-01160-f021:**
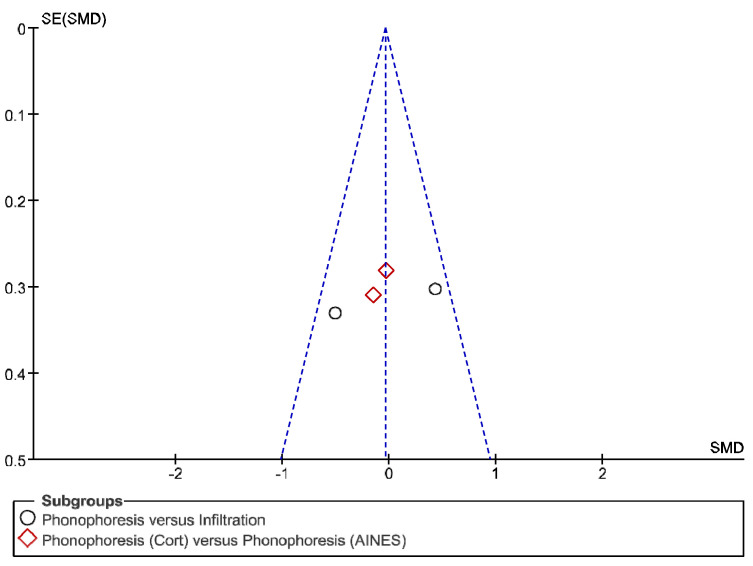
Phonophoresis for SNAPam.

**Figure 22 jpm-12-01160-f022:**
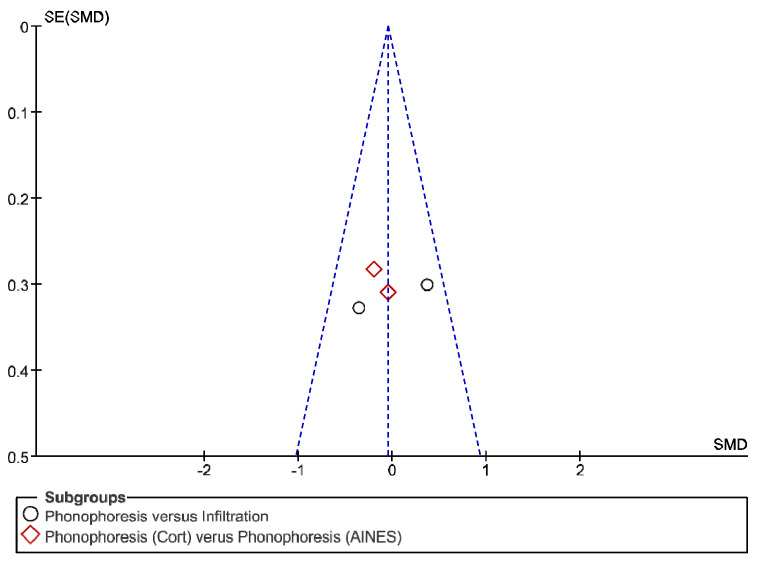
Phonophoresis for CNVM.

**Figure 23 jpm-12-01160-f023:**
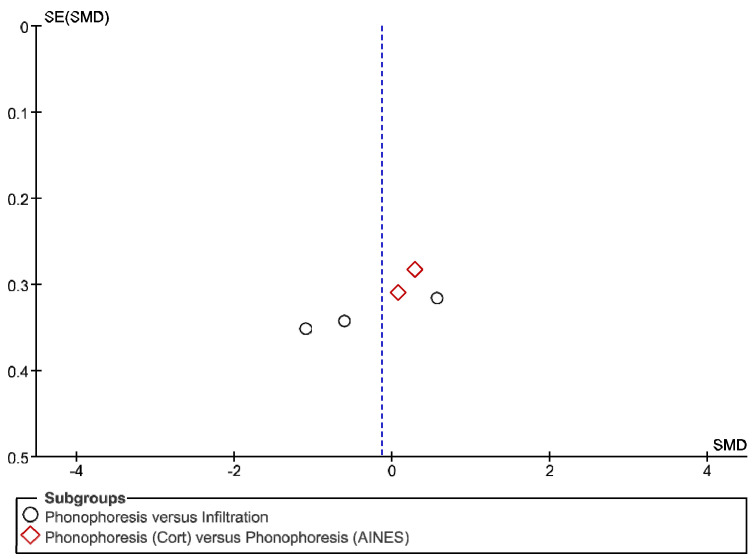
Phonophoresis for CNVS.

**Figure 24 jpm-12-01160-f024:**
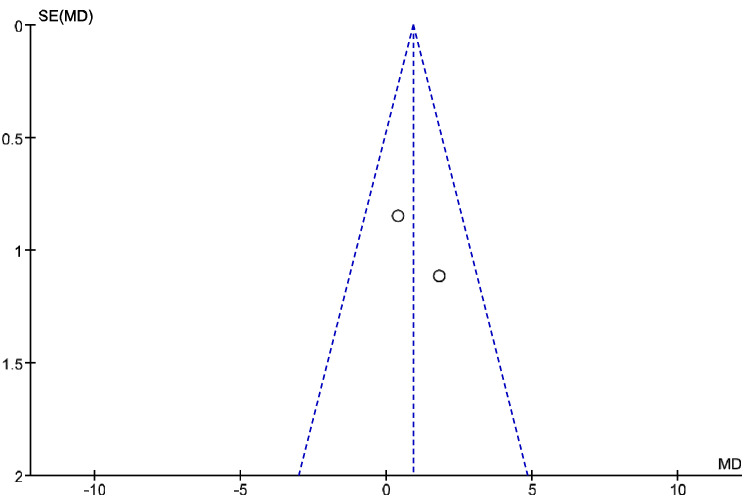
Phonophoresis for CMAPam.

**Figure 25 jpm-12-01160-f025:**
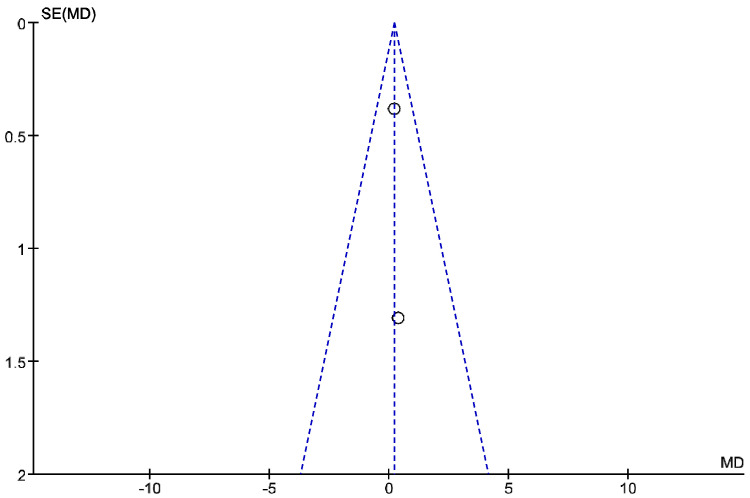
Phonophoresis for Pinchmeter.

**Table 1 jpm-12-01160-t001:** Search strategy in the databases used.

Search Formulae	Databases
Phonophoresis AND carpal tunnel syndrome Phonophoresis AND carpal tunnel syndrome AND treatment(s) Phonophoresis AND median neuropathy Phonophoresis AND median neuropathy AND treatment(s) Sonophoresis AND carpal tunnel syndrome Sonophoresis AND carpal tunnel syndrome AND treatment(s) Sonophoresis AND median neuropathy Sonophoresis AND median neuropathy AND treatment(s)	Web of Science
	SCOPUS
	PUBMED
	CINAHL Complete
	SciELO
“Phonophoresis” “carpal tunnel syndrome” “Phonophoresis” “carpal tunnel syndrome” “treatment(s)” “Phonophoresis” “median neuropathy” “Phonophoresis” “median neuropathy” “treatment(s)” “Sonophoresis” “carpal tunnel syndrome” “Sonophoresis” “carpal tunnel syndrome” “treatment(s)” “Sonophoresis” “median neuropathy” “Sonophoresis” “median neuropathy” “treatment(s)”	PEDro
	TOTAL

**Table 2 jpm-12-01160-t002:** Data abstracted from studies.

Authors (Year)/Design	Study Groups	Measuring and Evaluation Tools	Intervention	Parameters	Results
Asheghan M. et al., 2020 [22] RCT	No. = 42 (31 women. 36 right-handed). Average age: LCI: 48.6 (±11.6); LLLT: 49.4 (±5.2); PCS: 52.4 (±3.8) LCI: no = 14 (11 women) LLLT: no = 14 (9 women) PCS: no = 14 (11 women) CTS: Mild to moderate	Pain (VAS) BCTQ NCS Evaluation: Before- 4th week	LCI: methylprednisolone with lidocaine LLLT PCS: hydrocortisone acetate (10%)	LLLT: 10 sessions. (10 s/session) PCS: 10 sessions (3 times/week), Frequency: 1 MHz. Intensity: 1 W/cm^2^. ERA: 5 cm^2^. 5 min/session	All the three methods were effective. Statistically significant differences in terms of pain for LCI (*p* = 0.003) and for sensory latency (*p* = 0.001)
Boonhong J & Thienkul W. 2020 [32] RCT	No. = 33 (50 hands. 17 bilateral) Average age: 51.5 (±10.5) US: no = 16 hands PNSAI: no = 17 hands PCS: no = 17 hands CTS: Mild to moderate	BCTQ NCS Evaluation: Before 4th week	US: Contact gel PNSAI: Piroxicam (0.5%) PCS: Dexamethasone sodium phosphate (0.4%)	For all the treatments: 10 sessions (2–3 times/week) for four weeks. Continuous mode. Frequency: 1 MHz. Intensity: 1 W/cm^2^. 10 min/session	All the three methods improve clinical symptoms and functionality, but not the electrophysiologic parameters. Statistically significant differences are not observed among methods (*p* < 0.05)
Soyupek F. et al., 2012 [28] RCT	No. = 47 (74 hands. 28 bilateral. 14 right-handed. 4 left-handed) Average age: Splint: 47.9 (±6.9); PNSAI: 53.7 (±10.4); PCS: 50.5 (±8.7) Splint: no = 23 hands PNSAI: no = 23 hands PCS: no = 28 hands	Pain (VAS) Ecography (cross-sectional area of median nerve) Phalen and Tinel tests BCTQ NCS Evaluation: Before- 3 months	Splint: Neutral position PNSAI: diclofenac diethylammonium PCS: betamethasone valerate (0.1%)	PNSAI/PCS: 5 sessions/week for 3 weeks. Frequency: 3 MHZ. Intensity: 1.5 W/cm^2^. ERA: 5 cm^2^. 10 min/session	PCS proved more efficient, although no correlation was established between symptoms severity, functionality and ecographic and electrophysiologic findings. (*p* < 0.05)
Aygül R et al., 2005 [24] RCT	No. = 31 (56 hands. 31 women. 27 bilateral) Average age: LCI: 46 (±13.5); Ionto: 46.1 (±13.5); PCS: 44.1 (±5.7) LCI: no = 12 Ionto: no = 9 PCS: no = 10 CTS: Mild to moderate	BCTQ NCS Evaluation: Before 2 months & 3 months	LCI: dexamethasone sodium phosphate Ionto: dexamethasone sodium phosphate (0.1%) PCS: dexamethasone sodium phosphate (0.1%) for 3 weeks	Ionto: 3 weeks (5 days/week). Galvanic current from 1 to 4 mA. 10 min/session PCS: 3 weeks (5 days/week). Frequency: 3 MHz. Intensity: 1 W/cm^2^. ERA: 5 cm^2^. 10 min/session	LCI is more effective compared to electrophysiologic parameters. Statistically significant differences are not observed between PCS and Ionto. (*p* < 0.05)
Bakhtiary AH et al., 2013 [31] RCT	No. = 34 (52 hands. 18 bilateral. 16 right-handed). Average age: Ionto: 48.2(±14.5); PCS: 44.6 (±12.8) Ionto: no = 26 hands PCS: no = 26 hands CTS: Mild to moderate	Pain (VAS) Pinch and grip strength (Dynamometer) NCS Evaluation: Before-after and in week 4th	Ionto & PCS: dexamethasone sodium phosphate (0.4%)	Ionto: 2 mA/minute galvanic current. Dosage: 40 mA. 20 min PCS: 10 sessions (5 sessions/week). Pulsed emission (25%) Frequency: 1 MHz. Intensity: 1 W/cm^2^. 5 min/session	PCS is more efficient than Ionto (*p* < 0.05)
Soyupek F et al., 2012 [29] *(II)EC*	No. = 51 (84 hands. 33 bilateral) Average age: LCI: 51.34 (±10.18); PNSAI: 48.3 (±8.66); PCS: 49.24 (±12.27); Splint:47.52 (±8.36) Splint: 19 hands LCI: no = 23 hands PNSAI: no = 20 hands PCS: no = 22 hands	Pain (VAS) Grip strength (Dynamometer) Manual dexterity (The grooved pegboard) Semmes-Weinstein test Duruoz Hand Index Phalen and Tinel tests NCS Evaluation: Before- 3 months	Splint: Neutral position LCI: betamethasone dipropionate (0.5 mg) PNSAI: diclofenac diethylammonium (0.1%) PCS: betamethasone valerate (0.1%)	PNSAI/PCS: 3 weeks (5 sessions/week). Frequency: 3 MHz. Intensity: 1.5 W/cm^2^. ERA: 5 cm^2^. 10 min/session	For PCS improved NCS parameters are recorded, but not for pain and other subjective parameters (*p* < 0.05)
Gurkay E et al., 2012 [27] RCT	No. = 54 (45 right-handed. 7 left-handed) Average age: Splint: 43 (±6.9); Ionto: 44.1 (±9.5); PCS: 44 (±8.7) Splint: no = 18 hands Ionto: no = 16 hands PCS: no = 18 hands	BCTQ Grip strength (Dynamometer) Manual dexterity and function (Nine-holepeg test) Evaluation: Before- 3 months	Splint (all the groups): Neutral position Ionto: Betamethasone (0.1%) PCS: Betamethasone (0.1%)	Ionto: 3 weeks. (3 sessions/week). 4 mA galvanic current. 10 min/session PCS: 3 weeks. (3 sessions/week). Frequency: Continuous mode. 1 MHz. Intensity: 1 W/cm^2^. 10 min/session	All three methods were effective. Statistically significant difference in PCS BCTQ compared to splint. Variations are not observed concerning grip strength, manual dexterity, and function (*p* > 0.05)
Rüksen S et al., 2011 [23] RCT	No. = 32 (40 hands. 29 women. 9 bilateral) Average age: LCI: 41.3 (±11.2); PCS: 45.7 (±10.3) LCI: no = 20 hands (19 women) PCS: no = 20 hands (18 women) CTS: Mild to moderate	Pain (VAS) BCTQ Pinch and grip strength (Dynamometer) Paresthesia (Likert Scale) Manual dexterity (Test Grooved Pegboard) Evaluation: Before-after and in 3 months	LCI: (6.43 mg of betamethasone dipropionate) + splint + exercises PCS: (2.63 mg of betamethasone valerate) + splint + exercises	PCS: 2 weeks (5 sessions/week). Intensity: 1 W/cm^2^. 10 min/session	After treatment completion both methods recorded a statistically significant improvement. No statistically significant differences were observed in relation to the degree of efficacy of both treatments. (*p* < 0.05)
Tuncay R et al., 2005 [25] RCT	No. = 36 women Average age: LCI: 39.16 (±13.03) PCS: 44.05 (±8.73) LCI: no = 18 PCS: no = 18	BCTQ Pinch and grip strength (Dynamometer) Evaluation: Before- 3 months	LCI: (Betamethasone 1 mg) + splint in a neutral position at night PCS: (Betamethasone) + splint at night in a neutral position	PCS: 3 weeks. (3 sessions/week). Continuous mode. Frequency: 1 MHz. Intensity: 1 W/cm^2^. 10 min/session	Both methods were effective (*p* < 0.001). LCI improves nerve conduction velocity (*p* < 0.05)
Bakhtiary AH et al., 2014 [26] RCT	No. = 35 (51 hands) Ionto: no = 19 (25 hands) PCS: no = 16 (26 hands) CTS: Mild to moderate	Pain (VAS) Pinch and grip strength (Dynamometer) Paresthesia NCS Evaluation: Before-after and in week 4th	Ionto: dexamethasone sodium phosphate (0.4%) PCS: dexamethasone (0.4%)	Ionto: 2 weeks (1 session/week). 0.4 mA/cm^2^ continuous current. 10 min/session PCS: 2 weeks (1 session/week). Pulsed mode. Frequency: 1 MHz. Intensity: 1 W/cm^2^. 5 min/session	More efficacy of PCS. Improved grip strength (*p* = 0.006), reduced pain (*p* = 0.001) and improved NCS parameters (sensory: *p* = 0.001, motor: *p* = 0.008).
Dogan-Akcam F et al., 2012 [30] RCT	No. = 39 (69 hands. 30 bilateral) Average age: US simulation: 49.8 (±5.3); US: 46.2 (±12.1); PCS: 46.1 (±7.7) US simulation: no = 13 (24 hands) US: no = 13 (21 hands) PCS: no = 13 (21 hands)	Pain (VAS) BCTQ NCS Evaluation: Before- 2 weeks and 12 weeks	US: simulation (harmless contact gel) + exercises US: (harmless contact gel) + exercises PCS: (dexamethasone 0.1%) + exercises	For all the groups: 2 weeks (5 sessions/week). Intensity: 0.1 W/cm^2^ (except for US simulation: 0.0 W/cm^2^). 5 min/session	All the methods are effective in relation to clinical parameters and evaluations. PCS is more efficient and long-lasting compared to NCS parameters (*p* < 0.05)

No.: total number of subjects; no = number of subjects per group; LCI: local corticosteroid injection; CTS: carpal tunnel syndrome; LLLT: Low level laser therapy; PNSAI: phonophoresis with non-steroidal anti-inflammatory drugs; PCS: phonophoresis with corticosteorids; Ionto: Iontophoresis; US: ultrasounds; VAS: visual analogue scale; BCTQ: Boston Carpal Tunnel Questionnaire; NCS: nerve conduction studies.

**Table 3 jpm-12-01160-t003:** Studies methodological quality (PEDro).

Author (Year)	C1	C2	C3	C4	C5	C6	C7	C8	C9	C10	C11	TOTAL
Asheghan M. et al., 2020 [22]	-	1	1	1	1	0	1	1	1	1	1	9/10
Boonhong J & Thienkul W., 2020 [32]	-	1	1	1	1	0	1	1	1	1	1	9/10
Soyupek F. et al., 2012 [28]	-	1	0	1	1	0	1	1	0	1	1	7/10
Aygül R et al., 2005 [24]	-	1	0	1	1	0	0	1	1	1	1	7/10
Bakhtiary AH et al., 2013 [31]	-	1	1	1	1	0	1	1	1	1	1	9/10
Soyupek F et al., 2012 (II) [29]	-	0	0	1	1	0	1	1	0	1	1	6/10
Gurkay E et al., 2012 [27]	-	1	1	0	1	0	0	1	0	1	1	6/10
Rüksen S et al., 2011 [23]	-	1	0	1	1	0	0	1	1	1	1	7/10
Tuncay R et al., 2005 [25]	-	1	0	1	1	0	0	1	0	1	1	6/10
Bakhtiary AH et al., 2014 [26]	-	1	0	1	1	0	1	0	0	1	1	6/10
Dogan-Akcam F et al., 2012 [30]	-	1	1	1	1	0	0	1	1	1	1	8/10

C1: Eligibility criteria were specified. C2: Subjects were randomly allocated to groups. C3: Allocation was concealed. C4: Groups were similar at baseline regarding the most important prognostic indicators. C5: There was blinding of all subjects. C6: There was blinding of all therapists who administered the therapy. C7: There was blinding of all assessors who measured at least one key outcome. C8: Measures of at least one key outcome were obtained from more than 85% of the subjects initially allocated to groups. C9: All subjects for whom outcome measures were available received the treatment or control condition as allocated or, where this was not the case, data for at least one key outcome was analysed by “intention to treat”. C10: The results of between-group statistical comparisons are reported for at least one key outcome. C11: The study provides both point measures and measures of variability for at least one key outcome.

## Data Availability

Not applicable.

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
