# Peer review of "Use of Sonophoresis with Corticosteroids in Carpal Tunnel Syndrome: Systematic Review and Meta-Analysis"

_jpm, 2022, doi:10.3390/jpm12071160_

Round 1
Reviewer 1 Report
my recommendations for minor revisions:
Figure 1 – some information not in English but in Spanish language
Spelling - Phalen`s test instead of Phanel
Table 2 – 12 abbreviations that should be explained in the text and discussion or in the tables: LMMotor; LMSensor; SNAPam; CNVM; CNVS; CMAPam;
Author Response
Dear Editor and reviewers,
First of all, we would like to thank you for your comments and for allowing us to address the issues you raise to improve the manuscript’s quality. We appreciate your observations and the time devoted to the constructive criticism and feedback of our manuscript. Please find the answer to your comments below and the recommended changes have been highlighted in yellow in the manuscript.
Point 1: Figure 1: some information not in English but in Spanish language language
Response 1: The reviewer is indeed right. It has been corrected. Sorry for the mistake.
Point 2: Spelling - Phalen`s test instead of Phanel
Response 2: It has been corrected in page 6 in line 148. Sorry for the mistake.
Point 3: Table 2 – 12 abbreviations that should be explained in the text and discussion or in the tables: LMMotor; LMSensor; SNAPam; CNVM; CNVS; CMAPam;
Response 3: The acronyms are identified on the page 6, lines 142-145

Reviewer 2 Report
This manuscripts review and meta-analyze data concerning the use of corticosteroids applied by phonophoresis in carpal tunnel syndrome. I suggest the authors to use the journal template. Moreover, numbered lines are helpful to localize the issues. References must be in the appropriate citation style.
Introduction:
First line: two first sentences must be one: "The CTS is a neuropathic disease that occurs at..."
Line 7: a reference is needed to support this affirmation
Line 9: consider to replace "long/short run" by "long/short term"
Line 14: a reference is needed to support that the functional characteristics have been improved.
Study Justification: please remove this title, as is not an accepted section in this journal.
Objective: the same: is not an independent section from Introduction, just use a different paragraph (or not) with our titled "Objectives". By this, "Materials and methods" section will stay as number two, which is the appropriate considering journal's template.
Table 1: the number of papers collected cannot be in this section, as it is a Result and at this moment you have not explained the selection criteria yet. Remove from the table and supply this information in Results section.
Eligibility criteria: include that no language or date restriction was used.
Assessment of methodological quality: in the third line, replace "was" by "is" as this procedure of not counting the first criterion for score is the only way of using the scale. You also should rewrite the previous sentence as this point remains completely clear.
Figure 1: what is the meaning of "k"? Is not defined and the common sign to represent the number of studies or participants is "n". Once again, check the journal template. Finally, remove Spanish sentences in the figure.
In the fourth paragraph of results, authors explain the tools used for measurement and evaluation, but do not explain the outcomes measured. I suggest to explain each tool used (for example VAS) inside the outcome measured (pain in this case, function, quality of life or the correspondent in the others).
Study groups included in the meta-analysis (GLORIA): check table numbers as all of them are wrong. The first table in this section is titled "Table 2" but in the one over is already table 3. Nonetheless, the Forest Plot diagrams are not tables, they are in fact Figures. Please consider also to increase the quality of the Forest Plot pictures as the resolutions may be not enough for the publication.
Graphic 1-12: this journal do not uses "Graphics", only Tables and Figures.
Discussion: several paragraphs begins by "Concerning", which is quite repetitive. Please consider using "Regarding" or others.
Conclusion: conclusion must conclude according with the objectives of the work. You must conclude something, almost by saying that the application of corticosteroids by phonophoresis do not seem better that other treatments according with the results.
References: as I previously pointed, authors must follow the journal template and the reference style of the journal.
